# Tumor-Infiltrating Lymphocytes and PD-L1 Expression in Pleomorphic Lobular Breast Carcinoma

**DOI:** 10.3390/cancers15112894

**Published:** 2023-05-24

**Authors:** Menekse Göker, Stephanie Deblaere, Hannelore Denys, Glenn Vergauwen, Eline Naert, Liv Veldeman, Chris Monten, Rudy Van den Broecke, Jo Van Dorpe, Geert Braems, Koen Van de Vijver

**Affiliations:** 1Department of Gynaecology, Ghent University Hospital, 9000 Ghent, Belgium; stephanie.deblaere@uzgent.be (S.D.); glenn.vergauwen@uzgent.be (G.V.); rudy.vdb@skynet.be (R.V.d.B.); geert.braems@uzgent.be (G.B.); 2Cancer Research Institute Ghent (GRIG), Ghent University, 9000 Ghent, Belgium; hannelore.denys@uzgent.be (H.D.); eline.naert@uzgent.be (E.N.); liv.veldeman@uzgent.be (L.V.); chris.monten@uzgent.be (C.M.); jo.vandorpe@uzgent.be (J.V.D.); koen.vandevijver@uzgent.be (K.V.d.V.); 3Department of Medical Oncology, Ghent University Hospital, 9000 Ghent, Belgium; 4Department of Radiotherapy, Ghent University Hospital, 9000 Ghent, Belgium; 5Department of Pathology, Ghent University Hospital, Ghent University, 9000 Ghent, Belgium

**Keywords:** invasive lobular cancer, pleomorphic invasive lobular cancer, tumor-infiltrating lymphocytes, programmed cell death 1, programmed cell death ligand 1, 22C3 assay, SP142 assay, survival

## Abstract

**Simple Summary:**

The immunological profile of pleomorphic invasive lobular cancer is poorly investigated. pILC is characterized by more aggressive behavior and a worse prognosis; however, this rare subtype lacks a specific treatment approach. Here, we investigated the expression of sTILs and analyzed the PD-L1 expression levels from sixty-six patients with pILC. Moreover, we analyzed the association between sTILs and PD-L1 expression with other prognostic or predictive biomarkers and correlated sTILs and PD-L1 expression with survival outcomes. sTILS (≥1%) was present in 64% of the patients, and 36% of the tumors demonstrated a positive PD-L1 using SP142 (≥1%) and 28% had a positive PD-L1 score of ≥1 using 22C3. We found no differences between the molecular subtypes, the clinicopathological features, and the immune parameters, probably due to the small sample size of the HER2+ and TN subgroups. Larger trials on the immune composition of the subtypes of lobular breast cancer are needed.

**Abstract:**

Background: The prognostic and predictive role of stromal tumor-infiltrating lymphocytes (sTILs) is undetermined in pleomorphic invasive lobular cancer (pILC). The same applies for the expression of PD-1/PD-L1 in this rare breast cancer subtype. Here, we aimed to investigate the expression of sTILs and analyze the PD-L1 expression levels in pILC. Methods: Archival tissues from sixty-six patients with pILC were collected. The sTIL density was scored as a percentage of tumor area using the following cut-offs: 0%; <5%; 5–9%; and 10–50%. The PD-L1 expression was analyzed using IHC on formalin-fixed, paraffin-embedded tissue sections using SP142 and 22C3 antibodies. Results: A total of 82% of the sixty-six patients were hormone receptor positive and 8% of cases were triple negative (TN), while 10% showed human epidermal growth factor receptor 2 (HER2) amplification. sTILs (≥1%) were present in 64% of the study population. Using the SP142 antibody, 36% of tumors demonstrated a positive PD-L1 score of ≥1%, and using the 22C3 antibody, 28% had a positive PD-L1 score of ≥1. There was no correlation between sTILs or PD-L1 expression and tumor size, tumor grade, nodal status, expression of estrogen receptor (ER), or amplification of HER2. Our data did not show any difference in survival between the three molecular subtypes of pILC with respect to sTILs and PD-L1 expression. Conclusion: This study shows that pILCs show some degree of sTILs and PD-L1 expression; however, this was not associated with a survival improvement. Additional large trials are needed to understand immune infiltration in lobular cancer, especially in the pleomorphic subtype.

## 1. Introduction

Breast cancer is a highly heterogenous disease that can be caused by a variety of distinct genetic alterations in mammary glands, leading to clinical consequences. Invasive ductal cancer (IDC) is histopathologically the most frequent breast cancer, followed by invasive lobular cancer (ILC), the second most common histological subtype (~15% of all breast cancer cases) [1,2].

The immunological profile of invasive lobular cancer (ILC) is poorly investigated. These tumors are predominantly hormone receptor positive (estrogen receptor (ER) and/or progesterone receptor (PR)) with a low proliferation rate, and only a minority of ILCs express human epidermal growth receptor 2 (HER2) [3]. The current management of the majority of these tumors consists of surgery followed by endocrine therapy. Studies have demonstrated a lower response rate to chemotherapy compared with invasive ductal carcinoma of no special type (IDC-NST) [4,5]. Some studies documented a worse long-term survival in ILC patients compared with ER-positive IDC [6,7], and atypical metastatic sites, such as the peritoneum, the ovaries, and the gastro-intestinal tract are common [8]. Most ILCs have significantly lower levels of sTILs (a median percentage of 5% sTILs) compared with their ductal counterparts [9,10]. High sTIL levels in ILCs are associated with adverse prognostic factors and less favorable clinical outcomes [10]. In triple negative breast cancer (TNBC) and HER2-positive breast cancer (BC), a particularly strong correlation between sTILs, survival, and response to therapy have been demonstrated [11]. In early stage TNBC, the average value of sTILs was 23% (standard deviation 20%), and 77% of patients had 1% or more sTILs [12]. In triple negative and HER2-positive BCs, high sTIL levels are associated with a prolonged patient survival, a decreased distant recurrence, and an improved response to chemotherapy [11,13,14,15].

ILCs represent a heterogenous group of tumors, with the pleomorphic histologic subtype being a variant of ILC [16,17,18,19]. Morphologically, these lesions demonstrate a similar pattern of infiltrative growth to ILC, but exhibit a greater degree of pleomorphism (defined as larger cells with a marked nuclear pleomorphism, >4 times the size of lymphocytes or equivalent to the high-grade ductal carcinoma in situ, with or without apocrine features) and a higher mitotic count than classic ILC [2]. The immunological profile of pleomorphic invasive lobular cancer (pILC), which comprises approximately 1% of all BC cases, has not been investigated thoroughly. To the best of the authors’ knowledge, no data are currently available on sTIL expression in pILC.

One of the mechanisms of immune suppression involved in tumor progression is the programmed cell death 1 (PD-1)/programmed cell death ligand 1 (PD-L1) pathway [20]. In immune responses, the PD-1/PD-L1 pathway serves as a feedback mechanism to prevent excessive T cell activity and autoimmunity. In malignancy, PD-L1 upregulation counteracts the antitumor immune response and can prevent effective antitumor immunity. Tumoral PD-L1 expression is of considerable clinical interest due to the development of PD-1/PD-L1 blocking antibodies used for a variety of malignancies, which may improve the disease outcome [20]. PD-L1 expression is observed in approximately 20–40% of all BCs. Studies have demonstrated that PD-L1 expression varies across different BC subtypes, with positivity rates ranging from 0 to 83% [21,22]. A higher rate of PD-L1 expression has been demonstrated in a subset of BCs with aggressive clinicopathological and immunohistochemical tumor features, such as grade 3 tumors, ER- and PR-negative tumors, or tumors with a Ki67 proliferation greater than 14%, and this is associated with a lower overall survival (OS) [23,24]. A meta-analysis of 19,870 patients revealed a higher rate of PD-L1 expression in breast tumors with a pathological complete response (pCR) following neoadjuvant chemotherapy [23]. The currently available evidence consistently demonstrates a greater expression of PD-L1 in TNBC (61.4%), whereas its expression in ER-positive and HER2-positive tumors is lower. Previous studies have focused almost exclusively on IDC. However, available data suggest a lower PD-L1 expression rate in ILC compared with IDC (13% vs. 28%) [25]. Data on PD-L1 expression in pILC are scarce, a sole study by Dill et al. in 2017 has examined the PD-L1 status in this cohort of patients, reporting a positivity rate of 14.3% (2/14), determined with clone SP142 (threshold of ≥1%) [25].

Although pILC is characterized by more aggressive behavior and a worse prognosis, this rare subtype lacks a specific treatment approach. To date, there is no established routine assessment for sTILs and PD-L1 in pILC. In an effort to identify potential treatment methods, we performed an analysis of sTILs and the immunohistochemical expression of PD-L1 in a cohort of 66 pILC patients. For the latter, we compared two different IHC scoring methods: The 22C3 assay and the SP142 assay. Furthermore, we analyzed the association between sTILs and PD-L1 expression with other prognostic or predictive biomarkers and correlated sTILs and PD-L1 expression with survival outcomes.

## 2. Materials and Methods

### 2.1. Case Selection

Surgically resected primary breast cancers accrued over a 16-year period (2004–2020) were obtained from our pathology archives. Selection was performed by two independent pathologists (KVDV and JVD) based on the morphology and aberrant immunohistochemical E-cadherin staining at the time of diagnosis. Only cases with known ER, PR, and HER2 IHC results were included. All pathology slides and reports were reviewed, and only true pleomorphic invasive lobular cases were selected (*n* = 66). All patients were offered standard of care adjuvant therapy. The clinicopathological characteristics of the selected cases were extracted from the medical and pathology reports. The patient data include primary tumor characteristics (e.g., size, extent, grade, and hormone receptor status), nodal staging (number of nodes examined and number of involved nodes), the primary operation performed (lumpectomy vs. mastectomy), lymph node staging (sentinel node vs. axillary dissection), vital status, and survival. This study was approved by the Ethical Committee of Ghent University Hospital (BC-06646).

### 2.2. Quantification of sTILs

Hormone receptors and HER2 expression were used to classify tumors into the following phenotypes: ER+/HER2- (luminal A), ER-/PR-/HER2- (TNBC), and HER2+. The density of sTILs was reported as an overall percentage of the stromal area covered by immune cells (i.e., the area occupied by mononuclear inflammatory cells over the total intra-tumoral stromal area). The assessment of sTILs was performed on whole H&E slides of the primary tumor. Only resection specimens were used. sTILs were scored according to international guidelines (https://www.tilsinbreastcancer.org/, accessed on 25 April 2023) [26]. sTILs were evaluated within the borders of the invasive tumor. All mononuclear cells (including lymphocytes, histiocytes, and plasma cells) were scored, but polymorphonuclear leukocytes were excluded. Lymphocytic infiltration around normal lobules, DCIS, and blood vessels and in the previous biopsy site, areas of diathermy, or areas with crush artefacts were disregarded. The sTIL density scores were subdivided into the following categories: 0% (none); <5% (rare); 5–9% (mild); or 10–50% (moderate).

### 2.3. Immunohistochemistry

Formalin-fixed, paraffin-embedded (FFPE) tissue sections of 4-µm thickness were stained for PD-L1 proteins. Two diagnostic anti-PD-L1 clones (SP142 from Ventana and 22C3 from Dako Agilent) were used for PD-L1 assessment. Sections were counterstained with hematoxylin. PD-L1 labeling was blindly scored to patient characteristics. Perivascular TILs or TILs surrounding DCIS or normal breast lobules served as the internal control; due to the age of several FFPE tissue blocks, PD-L1 was determined to be non-assessable if a slide exhibited no staining at all.

The PD-L1 expression measured by the SP142 antibody was evaluated on tumor-infiltrating immune cells (ICs) and recorded as a percentage of the total number of ICs. PD-L1-positive tumor-infiltrating ICs were typically seen either as variably-sized aggregates located toward the periphery of the tumor mass, in stromal bands dissecting the tumor mass, or as single cells scattered in the stroma. The percentage of IC staining with anti-PD-L1 SP142 was scored as <1 or ≥1%.

The PD-L1 expression, as defined by immunostaining with anti-PD-L1 22C3, was determined by using the combined positivity score (CPS), which is the number of PD-L1 stained cells (tumor cells, lymphocytes, and macrophages) in the tumor divided by the total number of viable tumor cells and multiplied by 100. In our cohort, the cut-off values were determined as <1, ≥1, or ≥10.

### 2.4. Statistical Analysis

For the evaluation of the clinicopathological variables between the subgroups, a Fisher’s exact test was performed. An independent t-test was performed for continuous variables (age and tumor diameter). The associations between the numbers of sTILs and PD-L1 SP142 and between sTILs and PD-L1 22C3 immunohistochemical staining were assessed using Spearman’s rho correlation test.

The cumulative survival time was calculated using the Kaplan–Meier method and analyzed using the log-rank test. OS was calculated from the date of diagnosis to the date of death from any cause or the last follow-up date. Disease-free survival (DFS) was determined from the date of the initial diagnosis to the date of disease recurrence. All analyses were performed using SPSS version 28. A *p*-value of <0.05 (two-tailed) was considered statistically significant.

## 3. Results

### 3.1. Patient and Tumor Characteristics

The clinicopathological features of 66 patients with newly diagnosed pILC are summarized in Table 1. The median patient age was 59 years (range 35–84) and the median follow-up was 91 months (7.5 years). Of the 66 patients, most had tumors that were ER+ (82%); only 10% were HER2+ and 8% were triple negative (TN). All cases had grade 2 (45%) or 3 (55%) cytonuclear atypia. Most of the patients had stage II disease (50%) and 40% had node-positive disease. The mean tumor size was 34 mm. There was little difference in the features of the different subtypes, although there was a trend toward more aggressive features in TN and HER2+ tumors. Only the nodal staging was significantly different between the three subgroups, with the highest involvement of lymph nodes in TNBC. Six of the sixty-six patients developed distant metastases during the follow-up period: Five of them had a bone metastasis, four of them had a liver metastasis, and one patient developed a brain metastasis.

### 3.2. sTIL Distribution and PD-L1 Expression (22C3–SP142)

Subsequently, we assessed the density and composition of sTILs. In total, 64% of the study population demonstrated sTILs; 29% had less than 5% (≥1–4%), 17% had between 5 and 9%, and only 18% had 10% or more sTILs. No patient had a score of >50%. We also examined immune parameters by molecular subtype. Only twenty-three of patients had an sTIL score of >5%; three of seven patients in the HER2+ group and two of five patients in the TN group scored >5% compared to eighteen patients within the ER+/HER2- subgroup (50%).

Using the SP142 PD-L1 antibody to measure the percentage of ICs that express PD-L1, only 36% of tumors expressed PD-L1 in ICs when applying ≥1% as the cut-off. In the HER2+ subgroup, four of seven patients (57%) had a positive IC score of ≥1%, while two of five patients (40%) in the TN group had a positive IC score of ≥1% compared with fifteen patients in the ER+/HER2- (luminal) subgroup (32%).

The PD-L1 expression evaluated using the 22C3 antibody was graded as CPS < 1, ≥1, or ≥10. In 28% of the cases, the CPS was ≥1; three of seven patients in the HER2+ group (43%) and two of five patients in the TN group (40%) had a CPS of ≥1 compared with eleven patients in the ER+/HER2- subgroup (24%). Using this antibody, we found that only three patients had a score of ≥10, including two patients in the ER+/HER2- subgroup and one patient in the HER2+ subgroup (67% vs. 33%) (Table 2). For both immunoassays, there were no significant differences between the three subgroups, probably due to a quite small sample size.

### 3.3. sTIL Distribution and PD-L1 (22C3–SP142) Expression According to Patient Characteristics

There was no correlation in our cohort between the sTIL scores or PD-L1 expression scores with variables, such as age, tumor size, tumor grade, lymphovascular space invasion (LVSI), or nodal status (Table 3). We found no relationship between sTILs and PD-1/PD-L1 expression and the development of metastatis, probably due to the small number of metastatic patients. We noticed that the sTIL scores were higher in late stage and grade tumors, in lymph node positive tumors, in cases with LVSI, and in tumors with a Ki67 of ≥15%, but these findings were statistically not significant. These findings were also observed in analyses using anti-PD-L1 antibodies. There was a trend for PD-L1-positive tumors to show more aggressive features, such as a higher grade, stage, and proliferation index; LVSI; and nodal involvement.

### 3.4. Correlation of sTILs with PD-L1 Expression (22C3–SP142)

We assessed the relationship between the presence of sTILs and PD-L1 expression (Table 4). The PD-L1 scores were positive in 36% and 28% of the overall study population using SP142 and 22C3 antibodies, respectively. A total of 64% of tumors contained ≥1% sTILs and 35% of tumors showed ≥5% sTILs. Tumors with greater numbers of sTILs had a greater percentage of PD-L1-positive ICs; 47% of tumors with ≥10% sTILs displayed a positive PD-L1 SP142 score (ICs ≥ 1%); and 66% of tumors with ≥10% sTILs demonstrated a PD-L1 22C3 CPS of ≥10.

A strong positive correlation was found between the percentage of sTILs and the PD-L1 expression (SP142 and 22C3), as shown in Figure 1. The correlation was stronger between the percentage of sTILs and PD-L1 SP142 scores (r = 0.82, *p* < 0.001), than between the percentage of sTILs and PD-L1 22C3 scores (r = 0.67, *p* < 0.02).

### 3.5. Correlation between sTILs and PD-L1 (22C3–SP142) Expression and Patient Survival

We observed a significant difference in DFS between the three subtypes of pILC, observing a worse DFS for patients with TNBC (*p* = 0.004). However, the Kaplan–Meier survival curves did not demonstrate differences in survival between the tumor subtypes with respect to the presence of sTILs and PD-L1 expression, although a survival analysis was limited by the small sample size.

## 4. Discussion

The present study examines a dataset of surgically-treated pleomorphic lobular cancers, examining the presence of sTILs and the expression of PD-L1. The pleomorphic variant of lobular carcinoma is more likely to be ER- and HER2+ and has a poorer prognosis than classic lobular cancers [19]. This study is the first report examining the immune environment in pILC. Several important findings were revealed. First, we found that 64% of patients had at least 1% sTILs, with sTIL scores ranging between 1 and 30%. Second, 36% of the patients had a positive PD-L1 IC score of ≥1% using SP142, and 28% demonstrated a positive PD-L1 CPS of ≥1 using 22C3. In this cohort, we found no differences between the molecular subtypes, the clinicopathological features, and the immune parameters, probably due to the small sample size of the HER2+ (*n* = 7) and TN subgroups (*n* = 5). The only observed trend showed that higher sTIL scores and PD-L1-positive tumors might be associated with more aggressive morphological and clinical features, but this was statistically not significant.

Similar to invasive lobular breast cancers in general, we confirmed the lower levels of sTILs (mean 4.7%) compared to ductal carcinomas [9,27]. Recently, Tille et al. found a mean sTIL score of 2.7% in a cohort of 459 ILC patients (40.3% had an sTIL score of ≤5%, while only 7.6% had an sTIL score of >5%). sTILs were correlated with a higher grade, a larger tumor size, lymph node metastases, and the Nottingham prognostic index, suggesting a pro-tumorigenic role of sTILs in ILC [10]. Desmedt et al. demonstrated that lobular cancers with high sTIL levels were associated with a young age, lymph node involvement, and a high proliferation status as defined by Ki67 [9]. In addition, both studies found an association between histologic ILC variants and lymphocyte infiltration, as the alveolar histotype was found more often to be sTIL-negative, while mixed ILC carcinomas were mostly sTIL-enriched [9,10]. These findings suggest the role of sTILs in the poor outcome observed for patients with solid and mixed non-classic ILC versus patients with classical ILC [28].

High sTIL levels are associated with more favorable prognosis in TN and HER2+ tumors [12,15,29,30], whereas the prognostic role of sTILs in ER+ BCs is uncertain. To date, no study has demonstrated a favorable impact of sTILs on DFS, OS, or breast-cancer-specific survival in this molecular subtype [10,31,32,33,34]. Furthermore, in a recent meta-analysis of six neoadjuvant chemotherapy trials, higher sTIL scores were associated with a significant reduction in OS in luminal BCs, while an increase in the sTIL score was associated with longer DFS in TN and HER2+ BC [11]. However, these results must be interpreted with caution, since ER+ BCs have been divided into luminal A and luminal B tumors in recent decades, where the latter were associated with higher sTIL levels, a higher tumor mutational burden (TMB), and a worse clinical outcome than the luminal A subgroup [35]. In ILC, increasing sTIL levels are associated with a poor OS and a decreased DFS independent of lymph node metastases and ILC molecular subtypes, as determined in a multivariate analysis [10]. In our cohort, we could not identify sTIL expression as a factor associated with survival.

Breast tumors generally have a low TMB, are poorly infiltrated by sTILs, and have low levels of PD-L1; therefore, they are considered to be less responsive to immune checkpoint blockades (ICBs) compared to melanoma and lung cancer. However, the PD-L1 expression is significantly higher in invasive diseases compared to normal breast tissue [22,36], and many studies have found a positive correlation between PD-L1 expression and a more favorable prognosis [22,37,38,39], although others have found the inverse relationship [40,41]. Perhaps these contrasting results are related to different antibody choices, interpretations of immunohistochemical staining, cut-off points, evaluated cell types, and perceptions of positive cells in PD-L1 staining [21]. Dill et al. evaluated 245 primary and 40 metastatic (20 nodal and 20 distant metastasis) BCs with PD-L1 IHC on tissue microarrays and revealed that PD-L1 staining (clone SP142) of tumor cells was seen in 12% of primary BCs, including 31% of TNBCs. Tumor cell staining is common in ductal cancers with medullary (54%), apocrine (27%), and metaplastic features (40%). Staining for PD-L1 of ICs was observed in 29% of all BCs and in 61% of TNBCs [25]. The expression of PD-L1 in tumors cells and ICs is rare in ILCs (8.7 and 13%, respectively), while the pleomorphic subtype contained stromal PD-L1-positive immune cells in 14.3% of tumors (2/14). In contrast, Thompson et al. found that PD-L1 was expressed by tumor cells in 17% of ILCs (*n* = 47). In this study, the PD-L1 expression in stromal ICs was also assessed, with 71% displaying focal (<5%) or moderate (6–49%) labeling, and 29% displaying diffuse labeling (>50%) [42].

ER+ BC is characterized by a low PD-L1 expression, as found in a study by Sobral-Leite et al. [32]. In total, 410 primary, treatment-naïve breast tumors (162 ER+/HER2-, 101 HER2+, and 147 TNBC) were examined. At least 1% of PD-L1-positive tumor cells or ICs (determined with clone E1L3N) were observed in 53.1% of ER+/HER2-, 73.3% of HER2+, and 84.4% of TNBC tumors. Lower levels of sTILs and a lower PD-L1 expression were observed in lobular cancers compared to ductal cancers, regardless of other pathological variables [34,42]. We observed PD-L1 positivity in 36% of cases of pILCs using the SP142 IC score and in 28% of cases using the 22C3 CPS. We did not identify PD-L1 expression as a prognostic variable in our study; however, the HER2+ and TN groups were of limited sample size. In many studies, the PD-L1 expression in TN and HER2+ tumors was significantly associated with an improved survival. In addition, a strong positive correlation between PD-L1 positivity and sTIL density has previously been demonstrated [37,39].

PD-L1 expression patterns are different depending on the antibody clone used. SP142 detects more ICs but fewer tumor cells compared to other PD-L1 antibodies, and thus it is expected that SP142-positive BCs are enriched with sTILs, CD8+ cells, and other immune features [43,44]. Our study showed lower rates of PD-L1 positivity using the 22C3 CPS. Huang et al. compared three FDA-approved diagnostic immunohistochemistry assays for PD-L1 testing in a cohort of TNBC patients. They found that 5% of tumors were positive for PD-L1 using the SP142 IC score (1% cut-off), but negative when using 22C3 with the IC score (1% cut-off) or the CPS (CPS of 1 or 2), suggesting that 22C3 is not able to identify all tumors that test positive with SP142 using the IC score [45].

Despite the limited numbers of sTILs and the low levels of PD-L1 expression in lobular BC, there has been an effort to determine whether ICBs have a role in ER+ BCs. In heavily pretreated patients in a metastatic setting, the addition of pembrolizumab did not improve the median PFS [46,47]. In a neoadjuvant setting, the ISPY-2 trial treated 40 ER+/HER2- and 29 TNBC patients with pembrolizumab in combination with standard chemotherapy. In the ER+ subgroup, the addition of pembrolizumab yielded a higher rate of pCR compared to the chemotherapy arm (34% vs. 13%). As expected, improvements were observed in the TNBC cohort (60% vs. 20%) [48]. With these promising results, the ISPY-2 trial showed that by focusing on patients in the early setting, ICBs may have a beneficial role in ER+ diseases [49].

We recognize that this study has some limitations, mainly due to its retrospective nature and the low number of HER2- or TN pILCs. Second, we included FFPE tissue samples older than 3 years, which may influence the quality of antibody staining, leading to lower levels of PD-L1 expression. Notably, scoring sTILs on an H&E slide according to the guidelines of the International Immuno-Oncology Biomarker Working Group on Breast Cancer is always feasible, as no technical immunohistochemical staining is needed. Another limitation of our study is that we did not analyze the detailed proportion of CD4+ helper T cells, CD8+ cytotoxic T cells, FOXP3+ T regulatory cells, and CD68+ macrophages. Sobral-Leite et al. demonstrated that PD-L1 is expressed on multiple immune cells (CD68+ macrophages, CD4+, FOXP3+, and CD8+ T cells) in the breast tumor microenvironment, independent of the PD-L1 status of the tumor cells [34].

## 5. Conclusions

To conclude, the present study is the first to examine the active immune microenvironment in a large series of pleomorphic lobular carcinomas, evaluating sTILs and PD-L1 expression levels. We found that pILCs were characterized by low but variable levels of sTILs and PD-L1. We could not demonstrate that higher sTIL or PD-L1 levels were associated with prognosis. More research on the immune composition in the different subtypes of lobular breast cancer is needed to provide better insights into the immune composition of these tumors.

## Figures and Tables

**Figure 1 cancers-15-02894-f001:**
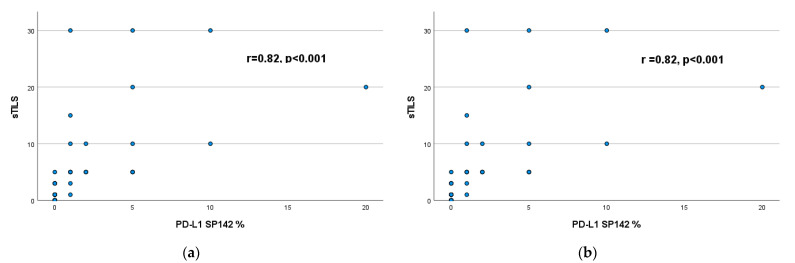
Correlation between (**a**) the percentage of sTILs and the PD-L1 SP142 score and (**b**) the percentage of sTILs and the PD-L1 22C3 score.

**Table 1 cancers-15-02894-t001:** Clinicopathological characteristics of the pleomorphic lobular cancer cohort.

	All Cases	ER+/HER2-	TNBC	HER2+	*p*
Total number *n* (%)	66	54 (82)	5 (8)	7 (10)	
Age (years)					0.61
Mean	58	58	61	62	
Median	59	59.5	57	62	
Tumor size *n* (%)					0.58
pT1	22 (33)	16 (30)	2 (40)	4 (57)	
pT2	33 (50)	29 (54)	2 (40)	2 (29)	
pT3	11 (17)	9 (6)	1 (20)	1 (14)	
Tumor size (mm)	34	35	42	26	
Grade (%)					0.17
G2	30 (45)	27 (50)	2 (40)	1 (15)	
G3	36 (55)	27 (50)	3 (60)	6 (85)	
LVSI *n* (%)					0.33
Negative	40 (61)	35 (65)	2 (40)	3 (43)	
Positive	26 (39)	19 (35)	3 (60)	4 (57)	
Lymph node metastases *n* (%)					0.014
pN0	38 (60)	31 (57)	1 (20)	6 (85)	
pN1	17 (25)	15 (28)	2 (40)	0	
pN2	7 (10)	7 (13)	0	0	
pN3	4 (5)	1 (2)	2 (40)	1 (15)	
Ki67 *n* (%)					0.21
<15	18 (37)	17 (42)	0 (0)	1 (16)	
≥15	31 (63)	23 (58)	3 (100)	5 (84)	
Missing	17				
Distant metastasis (%)					
Bone	5	3 (50)	1 (50)	1 (50)	0.3
Liver	4	2 (34)	1 (50)	1 (50)	
Brain	1	1 (16)	0	0	

Abbreviations: ER+: Estrogen receptor-positive; TNBC: Triple negative breast cancer; HER2+: Human epidermal growth factor receptor 2 positive; *n*: Number; LVSI: Lymphovascular space invasion.

**Table 2 cancers-15-02894-t002:** Immune parameters of the pleomorphic lobular cancer cohort.

	All Cases	ER+/HER2-	TNBC	HER2+	*p*
sTILs *n* (%)					0.26
0	24 (36)	18 (33)	3 (60)	3 (43)	
<5	19 (29)	18 (33)	0	1 (14)	
5–9	11 (17)	10 (18)	1 (20)	0	
≥10	12 (18)	8 (16)	1 (20)	3 (43)	
PD-L1 SP142 ICs *n* (%)					0.48
<1%	38 (64)	32 (68)	3 (60)	3 (43)	
≥1%	21 (36)	15 (32)	2 (40)	4 (57)	
Non-assessable	7				
PD-L1 22C3 CPS *n* (%)					0.58
<1	42 (72)	35 (76)	3 (60)	4 (57)	
≥1	13 (23)	9 (20)	2 (40)	2 (29)	
≥10	3 (5)	2 (4)	0	1 (14)	
Non-assessable	8				

Abbreviations: TNBC: Triple negative breast cancer; HER2+: Human epidermal growth factor receptor 2-positive; sTILs: Stromal tumor-infiltrating lymphocytes; PD-L1: Programmed death ligand 1; *n*: Number; ICs: Immune cells; CPS: Combined positivity score.

**Table 3 cancers-15-02894-t003:** Association between sTIL and PD-L1 scores and various clinical factors.

	sTILs				PD-L1 SP142		PD-L1 22C3		
	0%	1–4%	5–9%	≥10%	<1%	≥1%	<1	≥1	≥10
Age (years)									
Mean	57	61	57	58.5	57	60	58	56	66
Tumor (mm)	33	33	35	36	34	34.3	36	30	3
Stage, *n* (%)									
pT1	9 (37)	6 (32)	4 (36)	3 (25)	13 (34)	8 (38)	14 (33)	6 (46)	0
pT2	12 (50)	9 (47)	5 (46)	7 (58)	19 (50)	9 (43)	20 (48)	5 (38)	3 (100)
pT3	3 (13)	4 (21)	2 (18)	2 (17)	6 (16)	4 (19)	8 (19)	2 (15)	0
Grade, *n* (%)									
G2	12 (50)	9 (47)	3 (27)	6 (50)	19 (50)	6 (28)	18 (43)	5 (38)	1 (33)
G3	12 (50)	10 (53)	8 (73)	6 (50)	19 (50)	15 (72)	24 (57)	8 (62)	2 (67)
LVSI, *n* (%)									
Negative	16 (66)	11 (58)	7 (64)	6 (50)	24 (63)	11 (52)	27 (64)	5 (38)	2 (66)
Positive	8 (34)	8 (42)	4 (36)	6 (50)	14 (37)	10 (48)	15 (34)	8 (63)	1 (34)
Lymph node metastasis, *n* (%)									
pN0	16 (67)	10 (53)	6 (60)	5 (42)	22 (58)	11 (55)	25 (60)	5 (42)	2 (66)
pN1	4 (17)	6 (32)	3 (30)	4 (33)	9 (24)	6 (30)	10 (24)	4 (33)	1 (34)
pN2	3 (12)	2 (10)	1 (10)	1 (8)	5 (13)	1 (5)	5 (12)	1 (8)	0
pN3	1 (4)	1 (5)	0	2 (17)	2 (5)	2 (10)	2 (5)	2 (17)	0
HER2, *n* (%)									
Negative	21 (87)	18 (95)	11 (100)	9 (75)	35 (92)	17 (81)	38 (90)	11 (85)	2 (67)
Positive	3 (13)	1 (5)	0	3 (25)	3 (8)	4 (19)	4 (10)	2 (15)	1 (33)
Ki67, *n* (%)									
<15	7 (50)	7 (41)	2 (18)	2 (25)	13 (45)	2 (12) *	13 (41)	3 (18)	0
≥15	7 (50)	10 (59)	8 (82)	6 (75)	16 (55)	15 (88) *	19 (59)	9 (82)	3 (100)

Abbreviations: sTILs: Stromal tumor-infiltrating lymphocytes; PD-L1: Programmed death ligand 1; *n*: Number; LVSI: Lymphovascular space invasion; HER2+: Human epidermal growth factor receptor 2-positive; * *p*-value < 0.05.

**Table 4 cancers-15-02894-t004:** (a) Correlation between the percentage of sTILs and the PD-L1 SP142 score. (b) Correlation between the percentage of sTILs and the PD-L1 22C3 score.

**(a)**				
		**<1%**	**≥1%**	
		** *n* ** **(%)**	** *n* ** **(%)**	
sTILs	0%	21 (55)	0	
	<5%	16 (42)	2 (10)	
	5–9%	1 (3)	9 (43)	
	≥10%	0	10 (47)	
**(b)**				
		**<1**	**≥1**	**≥10**
		** *n* ** **(%)**	** *n* ** **(%)**	** *n* ** **(%)**
sTILs	0%	20 (48)	0	0
	<5%	16 (38)	2 (15)	0
	5–9%	4 (9)	5 (39)	1 (34)
	≥10%	2 (5)	6 (46)	2 (66)

Fisher’s exact test: *p* < 0.001.

## Data Availability

Data can be made public if desired.

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
