# Peer review of "Tumor-Infiltrating Lymphocytes and PD-L1 Expression in Pleomorphic Lobular Breast Carcinoma"

_cancers, 2023, doi:10.3390/cancers15112894_

Round 1
Reviewer 1 Report
This study performed an analysis of stromal tumor-infiltrating lymphocytes (sTILs) based on H&E staining and PD-L1 expression using antibody clones of 22C3 and SP142 in 66 cases with breast cancer subtype of pleomorphic invasive lobular cancer (pILC), and analyzed the association between sTILs and PD-L1 expression with clinicopathological parameters and with survival outcomes.
Although the results provide information on the rare pILC subtype according to the guideline of the International Immuno-Oncology Biomarker Working Group on Breast Cancer, the significance of H&E-stained total sTILs is not sufficient for the precise evaluation or explanation for therapy response and prognosis in patients with breast cancer. As a research article and limited case numbers, it would be better to examine the composition of sTILs to obtain more information for immune phenotype, because sTILs with different characteristics may exhibit opposite functions, such as CD8+ T vs Treg cells Th1 vs Th2 CD4 T cells, M1 vs M2 macrophages, and so on.
Section 2.2, ER+/HER2- should be defined as “luminal A” to distinguish from “luminal B” with HER2+
Table 2, variable “9-May” should be “5-9”.
Section 3.2, “18 patients within the ER+ /HER2- subgroup (78%)”, it should be 50%.
fine
Author Response
Please see also file in attachment.
We appreciate the time and effort that you have taken to provide feedback on our manuscript and we are grateful for the insightful comments on and valuable improvements to our paper. Please see below our response in red , and in blue, when we cited in the manuscript.
Comment: Although the results provide information on the rare pILC subtype according to the guideline of the International Immuno-Oncology Biomarker Working Group on Breast Cancer, the significance of H&E-stained total sTILs is not sufficient for the precise evaluation or explanation for therapy response and prognosis in patients with breast cancer. As a research article and limited case numbers, it would be better to examine the composition of sTILs to obtain more information for immune phenotype, because sTILs with different characteristics may exhibit opposite functions, such as CD8+ T vs Treg cells Th1 vs Th2 CD4 T cells, M1 vs M2 macrophages, and so on.
Answer: These are very interesting remarks. Sobral-Leite et al. published in 2018 the results of sTILS, PD-L1 and detailed multiplex immunofluorescent staining of the specific immune cells in a large series of different types of breast cancer. Our final author KVDV was one of the main pathologists in that paper. They demonstrated that a minor proportion of CD4+ helper T cells, CD8+ cytotoxic T cells, FOXP3+ T regulatory cells and CD68+ macrophages, co-express PD-L1, regardless of the pattern of immune infiltration (e.g. focal or diffuse/intratumoral PD-L1 +TILs). The data did not identify PD-L1 expression as a factor associated with survival in patients with ER+HER2− tumor. Furthermore, Sobral-Leite et al. did not find major differences/results of the phenotyping of sTILs in lobular cancers. In fact, those data on ILC were rather disappointing (personal communication/experience of KVDV), and we did not expect much results from the detailed analysis. On the other hand of course if we had more time and more additional funding, we would be able to extend this extra analysis.
We added this part in the discussion/limitation section (page 11, lines 366-370).
‘Another limitation of our study, is that we did not analyze the detailed proportion of CD4+ helper T cells, CD8+ cytotoxic T cells, FOXP3+ T regulatory cells and CD68+ macrophages. Sobral-Leite et al. demonstrated that PD-L1 is expressed on multiple immune cells (CD68+ macrophages, CD4+, FOXP3+, and CD8+Tcells) in the breast tumor microenvironment, independent of the PD-L1 status of the tumor cells [34].’

Reviewer 2 Report
The authors analyze the prognostic and predictive role of stromal tumor-infiltrating lymphocytes (sTILs) and PD-1/PD-L1 expression in 66 patients with pleomorphic invasive lobular cancer (pILC). They show that pILCs have some level of sTILs and PD-L1 expression, but this was not associated with a survival benefit. The manuscript is well written. The results are presented clearly and concisely. However, I have a questions on which the authors should comment:
1) pILC is a rare subtype with specific pathological features. I wonder if the authors could compare sTILs and PD-1/PD-L1 in pILC with their frequency in comparable invasive lobular carcinomas?
Author Response
Please see the attachment.
We appreciate the time and effort that you have taken to provide feedback on our manuscript and we are grateful for the insightful comments on and valuable improvements to our paper. Please see below in red our response.
Comment reviewer: pILC is a rare subtype with specific pathological features. I wonder if the authors could compare sTILs and PD-1/PD-L1 in pILC with their frequency in comparable invasive lobular carcinomas?
Answer: This is a very interesting remark. We agree that the addition of such information might improve the quality of our manuscript, however from the literature, we already know that invasive lobular cancers with classic pattern have significantly lower levels of sTILs (a median percentage of 5% sTILs) (Tille et al.2020, Desmedt et al.2018) and PD1/PD-L1 (Dill et al.2017). In 2018, Sobral-Leite et al. analyzed 410 primary treatment-naive breast tumors, including 33 ILC patients with lower expression of sTILs and PD-L1 (although PD-L1 was examined using another antibody, but with a lower cut off of 1% for TPS, IC and CPS). As final author KVDV was one of the main pathologists of the study of Sobral-Leite , he helped performing a specific immune cell typing of both ductal and lobular carcinomas using Multiplex Automated Image Acquisition and Analysis, next to sTILS and PD-L1. Partly due to the fact that classic pattern, luminal type ILC do not have high sTILs levels, the detailed multiplex IF stainings were rather disappointing and did not yield positive results. Repetition of a negative study did not seem useful, so in this manuscript, we focused only on pleomorphic lobular cancers because the immunological profile of these tumors has not been investigated thoroughly. Only the study of Dill et al in 2017 analyzed 14 patients with pleomorphic lobular cancer.

Reviewer 3 Report
To Author:
The pleomorphic lobular breast carcinoma (pILC) is a rare subtype of breast cancer. pILC has more aggressive behavior and worse prognosis. Meanwhile, pILC lacks a specific treatment approach. In this study, Menekse Göker et al. mainly analyzed tumor-infiltrating lymphocytes and PD-L1 expression in pleomorphic lobular breast carcinoma patients. However, the authors found no association between tumor-infiltrating lymphocytes / PD-L1 expression with tumor size, tumor grade, nodal status, and survival benefit. Meanwhile, the tumor-infiltrating lymphocytes have been extensively studied by others as breast cancer biomarkers (Carsten Denkert,Lancet Oncol. 2018; S Loi, Ann Oncol, 2021). Overall, this research lacks innovation and significance.
Comments:
(1) The infiltrating lymphocytes in breast cancer are heterogeneous, and they affect the treatment and prognosis of breast cancer. However, in this research, the authors did not consider the heterogeneity of infiltrating lymphocytes.
(2) In this research, the authors did not show the treatments these pILC patients received, which could affect the level of infiltrating lymphocytes in breast cancer.
(3) A key cause of death in cancer patients is tumor metastasis, and breast cancer can have brain metastasis, lung metastasis, and bone metastasis. However, in this research, the authors did not analyze the relationship between infiltrating lymphocytes and breast cancer metastasis.
Author Response
Please see the attachment.
The manuscript has been corrected by the language editing system of MDPI.

Reviewer 4 Report
In the present manuscript the authors have studied the presence of stromal TILs in pleomorphic lobular breast carcinoma and further analyzed the PD-L1 expression levels. They found that both some stromal TILs and PDL1 immunoreactivity were seen in this tumor histotype; however, no association between these features and improved survival was found.
The study is intriguing because it covers a debated topic as the relationship between TILs and PDL1 with lobular carcinoma.
The manuscript is also well written and both methods and results are clearly presented. I have some point that must be addressed:
1. I think that a brief introduction about the wide spectrum of breast tumors, including both epithelial and non-epithelial ones, would be useful for readers. Please see: PMID: 31965112
2. Please better specify the morphological criteria to diagnose an ILC and ILC, pleomorphic subtype in particular.
3. Please perform a minor english language editing of the manuscript.
Please perform a minor english language editing of the manuscript.
Author Response
We appreciate the time and effort that you have taken to provide feedback on our manuscript and we are grateful for the insightful comments on and valuable improvements to our paper. Please see below our response in red, and in blue, when we cited in the manuscript.
Comment 1: I think that a brief introduction about the wide spectrum of breast tumors, including both epithelial and non-epithelial ones, would be useful for readers. Please see: PMID: 31965112
We appreciate this comment. We have included (page 2, lines 45-49) a short introduction on the heterogeneity of breast cancer in the manuscript, with IDC and ILC the most common BC subtype.
‘Breast cancer is a highly heterogenous disease, that can be caused by a variety of distinct genetic alterations in mammary glands, implying clinical consequences. Invasive ductal cancer (IDC) is histopathologically the most frequent breast cancer, followed by invasive lobular cancer (ILC), the second most common histological subtype (~15% of all breast cancer cases) [1,2].
Comment 2: Please better specify the morphological criteria to diagnose an ILC and ILC, pleomorphic subtype in particular.
Thank you for this remark, we specified the morphological criteria better for the pleomorphic subtype using the WHO criteria in the manuscript, see below the text we added on page 2, lines 71-74 in the manuscript, and we included the reference of the WHO Classification of tumors [19].
‘Morphologically, these lesions demonstrate a similar pattern of infiltrative growth as ILC, but express a greater degree of pleomorphism (defined as larger cells with marked nuclear pleomorphism, > 4 times the size of lymphocytes /equivalent to that of high-grade ductal carcinoma in situ, with or without apocrine features); and a higher mitotic count than classic ILC. [19]’
Comment 3: Please perform a minor english language editing of the manuscript.
The manuscript has been corrected by the language editing system of MDPI.
Round 2
Reviewer 1 Report
The revised version is acceptable for publication.
Reviewer 3 Report
The authors revised this article very well this time. I don't have any comments.
Reviewer 4 Report
The manuscript has been improved. It is now acceptable.